# Physiological, Biochemical, and Transcriptomic Responses of *Neolamarckia cadamba* to Aluminum Stress

**DOI:** 10.3390/ijms21249624

**Published:** 2020-12-17

**Authors:** Baojia Dai, Chen Chen, Yi Liu, Lijun Liu, Mirza Faisal Qaseem, Jinxiang Wang, Huiling Li, Ai-Min Wu

**Affiliations:** 1State Key Laboratory for Conservation and Utilization of Subtropical Agro-Bioresources, Guangdong Key Laboratory for Innovative Development and Utilization of Forest Plant Germplasm, College of Forestry and Landscape Architecture, South China Agricultural University, Guangzhou 510642, China; baojiadai@outlook.com (B.D.); cc@cch3n.cn (C.C.); yiyiliuliu@outlook.com (Y.L.); faisal.ali522@gmail.com (M.F.Q.); 2Guangdong Key Laboratory for Innovative Development and Utilization of Forest Plant Germplasm, College of Forestry and Landscape Architectures, South China Agricultural University, Guangzhou 510642, China; 3State Forestry and Grassland Administration Key Laboratory of Silviculture in downstream areas of the Yellow River, College of Forestry, Shandong Agriculture University, Taian 271018, Shandong, China; lijunliu@sdau.edu.cn; 4Root Biology Center & College of Natural Resources and Environment, South China Agricultural University, Guangzhou 510642, China; jinxwang@scau.edu.cn; 5Guangdong Laboratory of Lingnan Modern Agriculture, Guangzhou 510642, China

**Keywords:** *Neolamarckia cadamba*, aluminum stress, signal transduction, hormone, ROS, DEGs

## Abstract

Aluminum is the most abundant metal of the Earth’s crust accounting for 7% of its mass, and release of toxic Al^3+^ in acid soils restricts plant growth. *Neolamarckia cadamba*, a fast-growing tree, only grows in tropical regions with acidic soils. In this study, *N. cadamba* was treated with high concentrations of aluminum under acidic condition (pH 4.5) to study its physiological, biochemical, and molecular response mechanisms against high aluminum stress. High aluminum concentration resulted in significant inhibition of root growth with time in *N. cadamba*. The concentration of Al^3+^ ions in the root tip increased significantly and the distribution of absorbed Al^3+^ was observed in the root tip after Al stress. Meanwhile, the concentration of Ca, Mg, Mn, and Fe was significantly decreased, but P concentration increased. Aluminum stress increased activities of antioxidant enzymes such as superoxide dismutase (SOD), catalase from micrococcus lysodeiktic (CAT), and peroxidase (POD) in the root tip, while the content of MDA was decreased. Transcriptome analysis showed 37,478 differential expression genes (DEGs) and 4096 GOs terms significantly associated with treatments. The expression of genes regulating aluminum transport and abscisic acid synthesis was significantly upregulated; however, the genes involved in auxin synthesis were downregulated. Of note, the transcripts of several key enzymes affecting lignin monomer synthesis in phenylalanine pathway were upregulated. Our results shed light on the physiological and molecular mechanisms of aluminum stress tolerance in *N. cadamba*.

## 1. Introduction

Aluminum is the third most abundant element followed by oxygen and silicon, and constitutes about 7% of shell mass [1]. The silicate and oxides of aluminum occurring in nature are nontoxic to plants; however, Al^3+^ ions released in acid soils are toxic to plant growth and development [2]. Acid soils account for 30-40% of the world’s arable land [3], accordingly aluminum toxicity has become an important limiting factor for crop growth and yield in acid soils [4,5,6]. Under acidic conditions, aluminum stress affects the physiological and molecular mechanisms of plants in many ways. The common phenomenon of aluminum toxicity on most plants is the inhibition of root growth [7,8,9], especially in distal transition zone of the root (DTZ) (1–2 mm) and elongation root zones (3–5 mm) from the root tip [10,11]. Furthermore, it also affects cell wall structure [12], as well as the physical and composition properties, osmotic kinetics, and structure of plasma membrane [13,14]. In addition, aluminum stress affects uptake of Ca^2+^ and other ions [15], results in oxidative stress [16,17], alters hormonal levels, initiates signal transduction [18], and changes cytoskeleton dynamics [19].

At cellular level, Al stress results in an imbalance of reactive oxygen species (ROS) [20,21,22], which induces changes in activities of protein kinases and transporters and affects the regulation of organic acid secretion and hormonal levels [23,24]. Importantly, aluminum stress can result in membrane lipid peroxidation, DNA damage, pigment degradation, cell wall thickening, and so on [25].

The cell wall is the outermost layer of the plant cell and is the first target of Al^3+^, and about 99% of the Al^3+^ absorbed by plants is deposited in the cell wall and cytosol [12,26,27], while the cell wall of the primary root (PR) mainly consists of pectin, cellulose, hemicellulose, polysaccharide, and protein [28,29,30]. In the cell wall of the PR, high Al^3+^ interferes with the binding of Ca^2+^ with cell wall pectin, affecting the pectin structures and the degradation of cell wall polysaccharides; thus, altering secondary cell wall structure and thickening the cell wall [31,32]. For example, xylan endoglucanase/hydrolase (XTH) and xylanase (XET), involved in xylan degradation, are limited by Al stress [26,33,34,35]. Likewise, pectin methyl esterase (PME) activity decreased and pectin was highly methylated in aluminum stressed rice [26,36], resulting in cell wall remodeling [37]. Aluminum toxicity also alters the contents of cell wall polysaccharides, for instance, aluminum stress in rice forces the Al rhizotoxicity1 complex (OsSTAR1) and rhizotoxicity2 (OsSTAR2) to transport UDP-glucose to the cell wall in the form of a bacterial-type ATP binding cassette (ABC) transporter to affect the cell wall structure [38]. Several studies have shown that aluminum affects the lignin content [39,40], which affects flexibility of cell wall. Of note, phenylalanine ammonia-lyase (PAL), 4-coumarate:CoA ligase (4CL), cinnamoyl-CoA reductase (CCR), caffeic acid *O*-methyltransferase (COMT), and cinnamyl alcohol dehydrogenase (CAD) are several key enzymes for the synthesis of lignin monomers from phenylalanine [41].

After entering the root tip cell wall, Al^3+^ interacts with anions on the plasma membrane surface to change the membrane potential and plasma membrane structure, thus changing the activities of ion transporters [42]. For example, a large number of Al^3+^ ions block the Ca^2+^ channel, resulting in a negative correlation between Al^3+^ and Ca^2+^ in calmodulin activity and other physiological activities [43], and competes with other cations for the binding sites on the surface of the membrane [44,45].

Al^3+^ is transported into cells either by the apoplastic or symplastic pathway and acts on organelles such as the nucleus, chloroplast, mitochondria, vacuoles, and lysosomes to induce aluminum tolerance genes, such as WRKY, MYB (v-myb avian myeloblastosis viral oncogene homolog), and bZIP (endosperm-specific basic Leu zipper) family genes, and induce transcription factors such as ERF (ethylene-responsive element binding factors), HSF (heat shock transcription factor), ARF (auxin response factor), NF-Y (nuclear factor Y), and the NAC (NAM, ATAF, and CUC) family [46,47,48,49], thus regulating their downstream gene expression [50]. For example, oxidative signal inducible 1 (OXI1) kinase mediates oxidative burst signal transduction to diverse downstream responses [51], glutathione S-transferases (GST) expression in *Arabidopsis thaliana* in response to oxidative stress, etc. [52]. Ferric chelate reductase 2 (FRO2) is an enzyme mediating the accumulation of reactive oxygen species [53]. In addition, NAC transcription factors are mainly involved in the process of senescence and apoptosis controlled by time evolutionary networks, and ANAC090 with ANAC017 plays a major regulatory role in salicylic acid SA and reactive oxygen species (ROS) responses [54,55]. Plants exposed to aluminum stress experience changes in the activities of superoxide dismutase (SOD), catalase from micrococcus lysodeiktic (CAT), malondialdehyde (MDA), peroxidase (POD), and other enzymes [16,20,56], which are closely related to the expression of genes involved in oxidative stress regulation. Some studies have shown that auxin influx OsAUX3-2 regulates root development [57], AtPIN3-shaped external current carriers mediate the transport of auxin [58,59], while the overexpression of OsPIN2, an auxin transporter gene, can reduce the production of aluminum-activated ROS [60]. HSF responds to abiotic stress as stress-induced and DNA-binding transcription factors. ERF is an ethylene-responsive factor involved in the regulation of plant hormones proteins regulate peroxidase [61]; the expression of these genes may be regulators of plant response to abiotic stress

The alterations in the cell membrane transport system are induced by altered expression of genes related to various element transporters, e.g., Al-induced secretion of malate (*ALMT*), multidrug and toxic compound extrusion (*MATE*) family, a bacterial-type ATP binding cassette (ABC) transporter, the zinc-finger protein (ZAT), and many others disturb the intracellular homeostasis of Al [38,62,63,64,65]. Studies have shown that external chelating and internal detoxification mechanisms are essential for tolerance against aluminum toxicity in higher plants [66,67]. The MATE family reduces aluminum toxicity via by chelating external Al^3+^ by regulating the activation and release of organic acids (citric acid, malic acid, and oxalic acid) to chelate external Al anions in roots [54,68], e.g., the mutation of WRKY46 results in downregulation of ALMT1 in *Arabidopsis* subjected to Al stress [69], while in *Sorghum bicolor (L.)* Moench, SbWRKY1 and DHHC type zinc finger transcription factor 1 (SbZNF1) trans-activated SbMATE [70]. ALMT1 and MATE1 mediate the exudation of malic acid and citric acid anions in wheat, soybean, and other crops [54,71,72]. The internal detoxification mechanism includes the regulation of Al^3+^ resistance transcription factor 1 (ART1) with downstream aluminum tolerance gene OsALS1 (tonoplast-localized Al transporter) [63,73] and sensitive to proton rhizotoxicity 1 in *Arabidopsis* (AtSTOP1) with downstream aluminum tolerance gene AtALS3. The expression of AtALMT1 and AtMATE1 [74,75,76] causes Al^3+^ to be blocked in vacuoles, while AtALS3 encodes ABC transporter [74], which makes Al^3+^ chelate with organic acid anions to form a stable and non-toxic complex to reduce aluminum absorption [75]. The heavy metal-associated isoprenylated plant protein 26 (HIPP26) may be involved in heavy metal transport [77,78]; this may also play a role in the process of aluminum stress.

*Neolamarckia cadamba* (*Roxb*.) Bosser is a large tree of the Neolamarckia subfamily of *Rubiaceae*. It belongs to tropical and subtropical tree species [79] and is mainly distributed in Guangdong, Guangxi, Yunnan, and other tropic regions in China and South Asia. It is a deep-rooted fast-growing tree species [80]. It has been widely planted and utilized due to its fast-growing, excellent wood properties, antibacterial, and anti-inflammatory effects. Red and yellow soil in southern China have the strongest acidity [81], so aluminum toxicity will affect the growth of *N. cadamba*, but the response mechanism of *N. cadamba* to aluminum toxicity has not been well studied, especially at physiological and molecular level. Therefore, in this study, we revealed the physiology and transcriptome responses of *N. cadamba* to aluminum stress and identified many differential expression genes (DEGs)

## 2. Results

### 2.1. Effects of Different Concentrations of AlCl_3_ on Growth of N. cadamba

Plant responses to AlCl_3_ stress differ with severity, tissue type, and duration of exposure. The response of *N. cadamba* to five different concentrations of AlCl_3_, i.e., 0 μM (CK), 50 μM, 100 μM, 200 Μm and 400 μM, was studied. Although no obvious changes in plant height and leaf growth were observed after 24 days of Al treatment, significant inhibition of root growth was recorded with increasing concentration of AlCl_3_ (Figure 1A). The root growth at 50 μM AlCl_3_ had no obvious difference when compared to 0 μM (CK, no Al) treatment. While the length of primary root (PR) at 400 μM AlCl_3_ for 4days was 0.13 cm, that at 0 μM (CK) reached 0.7 cm (Figure 1B). The inhibition percentage of PR under 400 μM AlCl_3_ treatment at 4days and 7days was 13.4% and 63%, respectively. After 7 days, the root net-growth tended to grow steadily. There was a non-significant effect of different Al concentrations on plant height as it changed from 6.6 cm (CK) to 6.1 cm (400 μM AlCl_3_ treatment) at 19 day (Figure 1C). To further study the effects of aluminum stress on the growth of *N. cadamba*, the biomass of *N. cadamba* at 7days treated with 400 μM AlCl_3_ was determined. In contrast to control plants, the aboveground dry weight and the belowground dry weight of *N. cadamba* seedlings in 400 μM AlCl_3_ decreased (Figure 1D,E), but this decrease was not statistically significant. This further indicated that AlCl_3_ could inhibit the root growth of *N. cadamba* under acidic conditions. Given that the effects of aluminum stress on growth of *N. cadamba* were mainly manifested by inhibition of root growth at early stage and increased severity with Al treatment, we further highlighted the changes in physiological and molecular mechanisms of *N. cadamba* treated with 400 μM AlCl_3_ for 0 days, 1 day, 3 days, and 7 days.

### 2.2. Inhibitory Effects of Aluminum Stress on N. cadamba Stem and Root Growth

To explore the effects of aluminum stress on *N. cadamba* growth, the cross section of the third node of the stem and 1 cm root tip of plants were examined. The secondary xylem of third node treated with 400 μM AlCl_3_ for 1 day, 3 days, and 7 days (Figure 2B,D,F) was thicker than that of the control plantlets (Figure 2A,C,E). At the same time, there was no obvious change in the secondary xylem of control roots between 0 days and 7th day (Figure 2G,H); however, the secondary xylem of the root tip was thickened on 7th day after 400 μM AlCl_3_ treatment (Figure 2I). This suggested that aluminum stress could increase the secondary xylem thickening of stem and root.

To further explore the inhibitory effects of aluminum stress on root growth, the whole root system of *N. cadamba* was scanned and analyzed after treatment with 400μM AlCl_3_ for 0 days, 1 day, 3 days, and 7 days, respectively. As shown, application of 400μM AlCl_3_ significantly inhibited total root length (Figure 2J), branch root number (Figure 2M), root cross number (Figure 2K), and root surface area (Figure 2L), but not root tip number (Figure 2N) and total root volume (Figure 2O) after 7 days treatment. These results indicate that high aluminum stress could not only inhibit the root elongation, but also affects root morphology.

### 2.3. Absorption and Distribution of Aluminum in Root Tip

Considering that aluminum stress inhibited the growth and development of *N. cadamba* roots, we explore the distribution of Al^3+^ in *N. cadamba* roots. We thus used staining and chemical methods to locate the distribution and concentration of aluminum and other elements. Compared with the control samples, the aluminum-treated 1 cm root tip was deep-red and brown, with more pronounced staining in the root cap region (Figure 3A). We thus speculated that the *N. cadamba*, after quickly absorbing Al^3+^, preferentially accumulates it in the root tip, which later diffuses to the whole root and then affects the root growth and is transported to stem and leaf.

High Al^3+^ ion concentration causes an ionic imbalance in the root system, thus resulting in altered concentration of a large number of other elements in root. Accordingly, we evaluated the quantity changes of aluminum (Al), calcium (Ca), magnesium (Mg), manganese (Mn), iron (Fe), potassium (K), phosphorus (P), and sulfur (S) in the root tip of *N. cadamba* after aluminum stress. The results showed that in control samples, the aluminum concentration in root tip reached to 180 μg/g at 7 days, while in the Al-treated plant there was a sharp increase in Al concentration at 1 day from 0 to 2140 μg/g and reached to 2631 μg/g at 3d. However, the accumulation rate slowed down and became almost constant with time extension (Figure 3B). In the control group, the Ca concentration increased to 220 mg/g from 0 days to 7 days (Figure 3C), while the Ca concentration in the treatment group decreased to 118 mg/g, 137 mg/g, and 85 mg/g on the 1st, 3rd, and 7th day, respectively. Our results showed that 400 μM AlCl_3_ treatment significantly increased the Al concentration and decreased the Ca concentration in *N. cadamba* on 1 day, 3 days, and 7 days.

Further analysis of other elements showed that aluminum stress affected the concentration of Mg, Mn, Fe, and P elements in the root tip of *N. cadamba*. Compared with the control, the concentrations of Mg (Figure 3D) and Mn (Figure 3E) in the root tip of *N. cadamba* treated with 400 μM AlCl_3_ for 1 day and 3 days were significantly decreased, but at the 7 days, they have same concentration. The Fe concentration (Figure 3F) was significantly decreased under aluminum stress at 3 days, and reached to 11 mg/g at 7 days. Likewise, the concentration of S element (Figure 3G) was significantly increased under aluminum stress for 1 day (*p* < 0.05), 3 days, and 7 days (*p* < 0.01), but not K (Figure 3H) and P (Figure 3I).

### 2.4. Effects of Al^3+^ Stress on Antioxidant Enzymes

In an acidic environment, aluminum stress can cause an accumulation of MDA, which leads to membrane lipid peroxidation, and increases SOD, POD, and CAT to maintain the stability of the cell membrane by scavenging ROS in an adverse environment [20]. Here, the MDA content and activities of SOD, CAT, and POD in leaves and root tips were determined after aluminum stress. The MDA content did not show obvious differences in the leaves (Figure 4A), but had an obvious decrease in root at 3 days and 7 days after treated with Al (Figure 4B), indicating that membrane peroxidation could occur in root after 3 days. Different from leaves (Figure 4C,E,G), the activities of SOD, CAT, and POD in roots were also increased from 1 day to 7 days (Figure 4D,F,H).

In addition, the chlorophyll content of *N. cadamba* after aluminum stress was quantified. The chlorophyll content decreased under aluminum stress from 1 day, 3 days, and 7 days, and the decrease in chlorophyll content under aluminum stress at 1 day was greater than 3 days and 7 days (Figure 4I). Aluminum stress can therefore cause rapid degradation of chloroplasts.

### 2.5. DEGs Shown in Aluminum Stress of N. cadamba by Transcriptome Sequencing Analysis

Aluminum stress mainly affected the growth and development of *N. cadamba* roots (Figure 1A); we thus studied the molecular mechanisms of *N. cadamba* roots in response to aluminum stress through RNA-seq. There was a total of 1,252,105,064 clean reads based on the quality evaluation as shown in Appendix A. The filtered sequences were analyzed for genome location, and the comparison between reads and the reference genome is shown in Appendix A. The reads alignment to the genome was more than 90%, and the match sequence of 85.88–92.16% was uniquely aligned to the reference genome, which satisfies the follow-up analysis. The Pearson correlation coefficient (R2) was greater than 0.92 correlation inspection index to test the correlation of gene expression levels between samples. The test results are as shown in Appendix A, which indicates that repeatability of the data under three biological replicates of are good enough.

To explore the mechanism of aluminum stress on *N. cadamba* at different times, genes from twenty-one libraries were assembled and six groups were compared (AL1_400 vs. AL1_CK, AL3_400 vs. AL3_CK, AL7_400 vs. AL7_CK, AL1_400 vs. AL0, AL3_400 vs. AL1_400, and AL7_400 vs. AL3_400). The volcanic map was established to determine the overall distribution of DEGs in response to aluminum stress at different time Appendix A, and, evaluated from the difference multiple and significant level, the result shows that a total 77,705 DEGs was selected from the six groups of comparison combinations, and the number of DEGs in the three horizontal comparison combinations of AL1_400 vs. AL1_CK, AL3_400 vs. AL3_CK, and AL7_400 vs. AL7_CK was much greater than that of the three vertical comparison combinations of AL1_400 vs. AL0, AL3_400 vs. AL1_400, and AL7_400 vs. AL3_400.

### 2.6. Verification of DEGs Through qRT-PCR

The RNA-seq sequencing data showed lots of DEGs that were significantly upregulated or downregulated after aluminum stress. To verify these DEGs, 18 DEGs relative expression patterns in three comparative combinations, i.e., AL1_400 vs. AL1_CK, AL3_400 vs. AL3_CK and AL7_400 vs AL7_CK were randomly selected for qRT-PCR detection (Figure 5). The corresponding primer pairs were listed in Appendix A. Of note, more than 94% of the gene expression patterns were consistent with RNA-seq sequencing results, so the results of RNA-seq sequencing data were reliable and repeatable. Next all DEGs were analyzed by principal component analysis (PCA). The obvious different abundance hierarchical clusters between the 0 days, 1 day, 3 days, and 7 days treatment groups and control group are shown in Figure 6A. This implies that aluminum treatment leads to the increase in total number of DEGs. Then, through the statistics of the difference of each gene in the root of *N. cadamba* under aluminum stress, there are 28,633 DEGs significantly expressed; the number of upregulated genes was significantly more than that of downregulated genes in the comparison between the treated group (R_400) and the control group (R_WT). However, the number of downregulated genes was significantly more than that of upregulated genes in the comparison combination between different time treatment groups (R_400) and 0 days control group (Figure 6B). Two Venn maps were constructed to analyze the common and specific DEGs in response to aluminum stress (Figure 6C,D). The results show that there were 1393 DEGs overlapping and stably expressed in the treatment groups and the control group; however, just 237 DEGs were expressed in the different time treatment groups and the control group.

### 2.7. Functional Enrichment Analysis of DEGs

Through gene annotation of sequence alignment with the model plant *Arabidopsis thaliana*, DGEs were shown in the following databases: GO (2405), NR (775), Swissprot (3058), KEGG (1193), and IPRSCAN (2405). In order to further analyze the function of DEGs during aluminum stress in *N. cadamba* root at different time intervals, we analyzed the DEGs by using the GO and KEGG databases. The enrichment GO terms (Appendix A) of DEGs to biological processes during aluminum stress for 1 day, 3 days, and 7 days include ion transport (GO:0006811), response to oxidative stress (GO:0006979), glucan metabolic process (GO:0006073, GO:0044042), and carbohydrate metabolic process (GO:0005975); the enriched GO term to cellular components include cytoplasm (GO:0044444, GO:0005737), cell wall (GO:0005618), and external assembly structure (GO:0030312); and the enriched GO term to molecular functions include ion transmembrane transporter activity (GO:0015075), inorganic molecular entity transmembrane transporter activity (GO:0015318), oxidoreductase activity, acting on paired donors, with incorporation or reduction of molecular oxygen (GO:0016705), and ion transmembrane transporter activity (GO:0015075).

The enriched KEGG analysis of all the DEGs of six differential combinations showed that a total of 416 metabolic pathways were involved in response of *N. cadamba* to aluminum stress at 1 day, 3 days, and 7 days. In this study, the related pathways of aluminum stress at different times ranked in the top 20 are shown in Figure 7. It was mainly enriched in phenylpropane biosynthesis and secondary metabolite biosynthesis pathway, followed by steroid biosynthesis, cysteine, and methionine metabolic pathway (FDR < 0.05). The enriched results of GO and KEGG showed that *N. cadamba* would adjust its metabolism to respond to aluminum stress. On the other hand, through the trend cluster analysis of the DEGs in different combinations after aluminum stress at different times, it is suggested that the DEGs of the treatment group and the control group may have similar functions or participate in the same metabolic process or cellular pathway together (Appendix A).

On the other hand, we also compared the three combinations of AL1_400 vs. AL0, AL3_400 vs. AL1_400, and AL7_400 vs. AL3_400 longitudinally to analyze the gene function clustering of *N. cadamba* after aluminum stress. We found that the key GO terms (Figure 8) enriched in the biological process include transmembrane transport (GO:0055085), transport (GO:0006810), response to stress (GO:0006950), etc. The molecular functions were mainly enriched in the cofactor binding, transition metal ion binding (GO:0048037, GO:0046914), and transcriptional regulation (GO:0016772, GO:0140110, GO:0003700). The cellular components were mainly enriched in membrane function (GO:0043231, GO:0044425, GO:0031224, GO:0016021, GO:0043227). The results revealed that there were significant differences in transcriptional regulation, metabolic processes, transport processes, and membrane functions induced by aluminum stress in different periods, which further indicated that the effects of aluminum stress on *N. cadamba* was significantly differed over time.

Comparative analysis with *Arabidopsis thaliana* genes showed that the relative expression of DEGs in the three combinations of AL1_400 vs. AL0, AL3_400 vs. AL1_400, and AL7_400 vs. AL3_400 (Appendix A), the expression of DEGs in different periods of aluminum stress was mainly downregulated by unknown functional gene *evm.TU.Contig491.32* acting on the membrane, unknown functional gene *evm.TU.Contig477.376*, and new gene *novel.619* responding to stress. The main upregulated genes in different periods were transcription factor NAC090 (*evm.TU.Contig488.87*), kinase *evm.TU.Contig108.3*, and enzyme FRO2 (*evm.TU.Contig477.127*), which can participate in heavy metal transport gene HIPP26 (*evm.TU.Contig262.36*), and some unknown functional genes were also upregulated.

### 2.8. Effects of Aluminum Stress on Cell Wall

The cell wall is the most important site of aluminum accumulation in Al stress. Al stress results in an increase in the lignin content. According to the RNA-seq data, the expression of 16 DEGs genes involved in the synthesis of phenylalanine lignin monomer induced by aluminum stress was screened (Figure 9A). Several key enzymes involved in the synthesis of lignin monomer were upregulated at different stages after aluminum stress. For instance, the expression of genes similar to *Arabidopsis thaliana* PALA (*evm.TU.Contig766.86*), CCR1 (*evm.TU.Contig45.284*),CAD (*evm.TU.Contig 462.293*), and 4CL1 (*evm.TU.Contig600.113*) were significantly upregulated at 1 day, 3 days, and 7 days of aluminum stress. The proportion of upregulated genes involved in phenylalanine biosynthesis was higher in different periods, indicating that in *N. cadamba* root cell wall thickness and decreased cell wall malleability were regulated by altered lignin synthesis during aluminum stress.

In this study, it was found that aluminum stress not only decreased the extensibility of the cell wall of root tip, but also decreased the remodeling of the cell wall. The synergistic effect of these two aspects led to inhibition of root growth. By calculating the FPKM values of different combinations under different treatment times (Figure 9B–D), we found that the expression of XTH family genes (*evm.TU.Contig63.448*, *evm.TU.Contig180.195*, *evm.TU.Contig553.99*, *evm.TU.Contig28.439*, *evm.TU.Contig180.196*, and *evm.TU.Contig35.15*) were significantly downregulated; moreover, the demethylation and esterification of cell wall pectin, regulated by PME gene family (*evm.TU.Contig284.2*, *evm.TU.Contig612.63*, and *evm.TU.Contig600.380*), was decreased significantly at 1 day, 3 days, and 7 days during aluminum stress.

### 2.9. Transport-Related Genes Respond to Aluminum Stress

As shown in Figure 10A, aluminum stress induced the expression of transporters and protein kinases in *N. cadamba*. Importantly, among DEGs, transporter-encoding genes such as ABC, ALMT, MATE, and ZAT families are enriched. For example, the *evm.TU.Contig46.7*, *evm.TU.Contig7.260*, and *evm.TU.Contig21.352* genes encoding the bacterial ABC transporter family were significantly upregulated. The ALMT family genes (*evm.TU.Contig154.214*, *evm.TU.Contig154.133*, *evm.TU.Contig488.39_evm.TU.Contig488.40*, and *evm.TU.Contig488.38*), MATE family genes (*evm.TU.Contig490.37*, *evm.TU.Contig81.157*, and *evm.TU.Contig66.895*), and *evm.TU.Contig63.553* similar to *Arabidopsis thaliana* AtALS1 genes involved in organic acid regulation were significantly upregulated at different stages of aluminum stress, suggesting that *N. cadamba* can tolerate aluminum stress on roots by increased secretion of organic acids. In addition, the expressions of *evm.TU.Contig447.151*, *evm.TU.Contig765.14*, *evm.TU.Contig14.131*, and *evm.TU.Contig421.426* genes, homologs of the of *Arabidopsis thaliana* sugar transporter ERD family, were downregulated during aluminum stress, indicating that *N. cadamba* could modulate its glucose metabolism and root nutrient absorption in response to aluminum stress. Additionally, the uptake and transport of other elements were affected by aluminum uptake. Together, aluminum stress altered the expression of different transporters. For example, the expression of intracellular calcium transduction signal gene CCR1 (*evm.TU.Contig63.493*) and magnesium transport protein regulatory gene *evm.TU.Contig16.298* was significantly downregulated. These results are in accordance with the decreased concentration of calcium and magnesium (Figure 3C,D). Although the potassium concentration has not changed in treatment, the expression of POT genes (*evm.TU.Contig41.8*, *evm.TU.Contig341.538*) regulating potassium transporter family was upregulated during aluminum stress.

The aluminum stress-induced transporters may conduct the signals through the corresponding protein kinases. The data analysis showed that aluminum stress could significantly affect expressions of some transport-related proteins and protein kinase regulatory genes (Appendix A) the higher expression of these genes may also raise aluminum absorption and transport. In addition, the expression of auxin steady-state protein kinase TMK1 (*evm.TU.Contig298.91*), auxin signal transduction kinase PKS1 (*evm.TU.Contig184.404*), and polar developmental receptor-like kinase PXL family genes (*evm.TU.Contig81.784*, *evm.TU.Contig855.26*, *evm.TU.Contig341.255*) were significantly downregulated at different stages of aluminum stress. However, the expression of *evm.TU.Contig256.319* and *evm.TU.Contig108.2* that may be involved in the abscission process and oxidative burst transduction kinase OXI1 (*evm.TU.Contig421.76*) was significantly upregulated.

### 2.10. Response of Transcription Factors to Aluminum Stress

Transcription factors play important roles on the stress tolerances in higher plants. Our RNA-seq showed that some ERF and WRKY transcription factors were significantly upregulated in the root tip under aluminum stress (Figure 10B). The expressions of these genes were similar to *Arabidopsis thaliana* WRKY23 (*evm.TU.Contig28.225*), WRKY33 (*evm.TU.Contig462.255*), WRKY7 (*evm.TU.Contig66.622*), WRKY70 (*evm.TU.Contig96.340*), and so on were significantly upregulated, and the expression of ethylene-responsive transcription factor ERF family genes were upregulated during aluminum stress. The expression of BZIP60 (*evm.TU.Contig14.384*), a zinc finger transcriptional factor, was significantly upregulated under aluminum stress, indicating that WRKY, ERF, and ZIP family genes were involved in downstream genes network in response to aluminum stress. HSFB2A (*evm.TU.Contig360.56*, *evm.TU.Contig35.177*) and HSFA4B (*evm.TU.Contig708.60*) genes were significantly upregulated (Figure 10B), indicating that *N. cadamba* may induce heat shock HSF family genes also participating aluminum stress response pathway.

### 2.11. Hormone-Related Genes Respond to Aluminum Stress

Plant hormones are key to plant growth and adaptation to various stress conditions; the expression level of hormone-related genes induced by aluminum stress in the root tip are shown in Figure 11A. Aluminum stress significantly downregulated the expression of auxin response factor (ARF) *evm.TU.Contig21.408* in auxin response and downregulated the expression of downstream genes IAA7 (*evm.TU.Contig447.342*, *evm.TU.Contig78.7*) and similar genes LAX3 (*evm.TU.Contig268.2*) and PIN3 (*evm.TU.Contig371.57*) in *Arabidopsis thaliana* after 1 day, 3 days, and 7 days aluminum stress. The expression of SAUR36 (*evm.TU.Contig256.89*, *evm.TU.Contig797.202*) was upregulated, the downregulation of auxin-regulated genes may regulate the down-regulation of auxin receptor protein gene ABP20 (*evm.TU.Contig600.242*). This suggests that regulating the expression of auxin-related genes might be a key factor in inhibition of root tip growth by aluminum stress.

Ethylene is a key hormone that inhibits plant growth and development. In the present study, aluminum stress significantly upregulated expression of ERF family genes (*evm.TU.Contig948.4*, *evm.TU.Contig151.18*, *evm.TU.Contig533.8*, *evm.TU.Contig84.515*, and *evm.TU.Contig84.397*, *evm.TU.Contig96.392*). Meanwhile, expression of abscisic acid receptor gene PYL4 (*evm.TU.Contig84.258* and *evm.TU.Contig462.238*) was significantly upregulated, while gibberellin-regulated genes GASA9 (*evm.TU.Contig81.65*) and GASA14 (*evm.TU.Contig66.216*), and cytokinin-regulated genes LOG8 (*evm.TU.Contig381.14*) and CKX7 (*evm.TU.Contig96.52*) were significantly downregulated. The significant expression of hormone-related genes indicates that aluminum stress affects *N. cadamba* root growth via regulating the transcription of hormones biosynthesis and or signaling-related genes.

### 2.12. ROS-Related Genes Respond to Aluminum Stress

Under low pH conditions, aluminum stress can increase the level of ROS, causing membrane lipid peroxidation and inducing the expression of genes related to oxidative stress. Aluminum stress significantly increased the expression of OXI1 (*evm.TU.Contig421.76*) in *N. cadamba* and *Arabidopsis thaliana* to mediate oxidative burst. RNA-seq analysis showed that PER and GST family genes were significantly expressed in aluminum stress-induced parameters and oxidative stress response (Figure 11B). This may be the main regulatory factor of oxidative stress induced by aluminum stress. The expression of PER family genes in *N. cadamba* was significantly downregulated at 1 day, 3 days, and 7 days after aluminum stress, which was consistent with the decreasing trend of POD activity at 3 days and 7 days. The downregulation of PER family and GST family gene expression after aluminum stress indicates that aluminum stress resulted in the reduction of enzyme activity, which led to the increase in H_2_O_2_ content, the decrease of binding ability of glutathione to various electronic compounds, and an increase in ROS content that led to imbalance of metabolic level. However, at the same time, the transcripts of *GSTU9* (*evm.TU.Contig130.57*) were significantly upregulated. It is probable that *evm.TU.Contig130.57* is involved in detoxification and antioxidation of osmotic stress caused by ROS in order to avoid adverse effects of aluminum stress on *N. cadamba* growth.

## 3. Discussion

In this study, hydroponic culture was used to study the effects of AlCl_3_ treatment on *N. cadamba* in acidic environment (pH4.5) with simple calcium solution. We revealed the responses of *N. cadamba* to aluminum stress at the physiological and molecular levels through an RNA-seq approach. Numberous DEGs have been uncovered when subjected to Al stress in *N. cadamba* roots at different time points. Our results provide useful cues to plant biologists.

In line with previous studies, the most rapid and obvious effect of aluminum stress on *N. cadamba* is the inhibition of root growth (Figure 1A). Our results are consistent with the phenotype of Gramineae plants, but they have a significant concentration difference and action time difference with Gramineae plants. As reported, the inhibition rate of root elongation of japonica rice variety (Koshihikari) reached 42% after being treated with 50 μM aluminum for 24 h, while that of indica rice variety (Kasalath) reached 73% [82]. The root elongation of wheat line Atlas66 (aluminum-tolerant genotype) decreased from 57.5% to 18.71% with an increase of aluminum concentration from 50 μM to 100 μM, and that of EM12 (the main fine variety in China) decreased from 30.09% to 3.09%. The change of Scout66 (aluminum sensitive genotype) decreased from 19.02% to 0.97% [83], while in Solanaceae, 700mM AlCl_3_ significantly decreased the relative root growth of *Actinidia chinensis Planch* without time dependency [84], and 250–500 mM AlCl_3_ treatment significantly inhibited the growth of lateral roots of chickpea. Moreover, the rooting pattern was affected by AlCl_3_ concentration gradient [85]. Interestingly, 50 μM AlCl_3_ had a certain promoting effect on the root growth, while 100 μM, 200 μM, and 400 μM AlCl_3_ concentrations inhibited the root elongation and root morphology of *N. cadamba*, and the inhibitory effect was more significant with the increase of concentration.

Aluminum stress induced the expression of the XTH gene family with tissue specificity by targeting the cell wall. OsXTH1 is only expressed in rice roots, but AtXTH1 is expressed in anthers tissue of *Arabidopsis* [35,86]. AtXTH26 is specifically expressed in roots [87], AtXTH32 is specifically expressed in stems. The XTH family genes in *N. cadamba* mainly showed the downregulation of XTH1, XTH8, XTH32, and XTH26 in the root tip to deal with aluminum stress. There are obvious differences between *Arabidopsis thaliana* and rice, so it is necessary to further determine the XTH in other tissues of *N. cadamba*.

The interaction of Al^3+^ with other ions has been revealed [88]. *N. cadamba* showed obvious differences in element absorption under aluminum stress (Figure 3), including the increase in Al and the invariance of K. To our surprise, the K concentration in grasses such as rice, corn, and wheat under aluminum stress changes significantly [88,89,90,91]. However, our transcriptome sequencing (RNA-seq) indicates that the expression of K transporter genes *evm.TU.Contig41.8* and *evm.TU.Contig341.538* were significantly upregulated at 1 day, 3 days, and 7 days of aluminum stress, However, the concentration of K in the root tip did not change, which may be due to the transfer of potassium absorbed by *N. cadamba* to other parts (Figure 3H and Figure 10A). This means that although the ion transporters in *N. cadamba* were significantly affected under aluminum stress, the ion levels were balanced at the same time to ensure normal growth. In this study, aluminum stress generates a very significant change in the levels of cytoplasmic Al^3+^ and Ca^2+^ in the root system. We found that a large amount of Al^3+^ absorbed by the root tip was found to be mainly distributed in the root tip (Figure 3A). This indicates that *N. cadamba* copes with Al stress through exclusion of Al in cell wall. Additionally, the concentrations of Mg and Mn in the root tip of *N. cadamba* decreased significantly after 1 day and 3 days of aluminum stress, but there was no significant change compared with the control at 7 days, which may be due to the absorption of self-regulating elements after adaptation to aluminum stress. At the same time, the downregulation of *evm.TU.Contig16.298* by aluminum stress may also be one of the reasons for the significant decrease of magnesium concentration, which is consistent with previous study [89]. Thus, it is suggested that Al^3+^ may potentially compete with other ions for enzyme binding sites and interfere with the absorption, migration, and utilization of other elements, thus affecting the nutritional imbalance at the overall plant level or in individual cells.

In addition to the element absorption, the tissue-specific changes caused by Al stress that change the cell membrane are also reflected in the activities of related enzymes that quench ROS. Past studies on wheat indicate that Al stress increases the activities of SOD and CAT enzymes [92], but both of them are inhibited in microalgae *Scenedesmus* spp. [56]. In addition, in this study, aluminum stress boosts the activities of SOD, CAT, and POD in the root tip of *N. cadamba*, while the content of MDA decreased. Surprisingly, MDA, which is one of the indicators of ROS content, increased in rice under aluminum stress [20]. This means that there are differences and similarities in the change trend of ROS in different plants under aluminum stress. In addition, changes of ROS content in leaves and roots also indicated that there are certain differences among different tissues.

Under acidic conditions, the growth and development of *N. cadamba* was limited by aluminum stress (Figure 2), which was closely related to the imbalance of hormone expression, the decrease of cell wall malleability, the damage of the cell membrane, and the expression regulation of genes related to ROS changes. RNA-seq analysis showed that *N. cadamba* mainly promoted the expression of ethylene regulatory genes, inhibited the expression of auxin-related genes, and then regulated the expression of ROS and lignin-related genes in response to aluminum stress. At the same time, the action of Al^3+^ on the cell membrane limits the absorption, transport, and utilization of other elements, which also induces changes in the activity of related ion transporters, further regulating and supplementing the necessary element balance for plant growth. In addition, a large number of Al^3+^ ions can induce the expression of related protein kinase-regulated genes, and then make the corresponding transcription factors respond to aluminum stress by regulating the expression of organic acid secretion-related transporter genes or ROS and lignin monomer synthase-related genes. We summarize the possible molecular regulatory mechanisms in Figure 12. However, we also found that some new genes or functional unknown genes were significantly expressed after aluminum stress (Appendix A); the specific mechanism of these novel genes requires further exploration.

## 4. Materials and Methods

### 4.1. Materials

Using the existing tissue culture seedlings of *N. cadamba* in State Key Laboratory for Conservation and Utilization of Subtropical Agro-Bioresources, the excellent single strain was selected for tissue culture and propagation, and the tissue culture seedlings with the same growth state were selected as experimental materials for aluminum stress water culture treatment.

### 4.2. Seedling Culture

At about 15 day (d) of tissue culture seedlings were domesticated for 4-7 days, and the seedlings with plant height of about 4–5 cm were cultured in water in the greenhouse with light of 5000 luxe and room temperature of 25 °C, with a light/dark cycle of 13 h/11 h. First, cultured in water for 6–8 days, then incubated in Hoagland solution [93] for 3–4 days, then cultured in deionized water for 3 days. Then the uniform seedlings were transferred into aluminum-containing nutrient solutions (pH4.5) supplemented with calcium (CaCl_2_ 0.5mM). The nutrient solution was changed every 3 days, oxygen pump was added to each culture box. The samples were sampled as described, and the related physiological and molecular parameters were determined as described.

### 4.3. Experimental Design

In order to screen the most obvious phenotype of *N. cadamba* under aluminum stress, five AlCl_3_ concentration gradients were set under the acidic condition of pH 4.5, which were 0 μM/L, 50 μM/L, 100 μM/L, 200 μM/L, and 400 μM/L. Each biological repeat contained five technical repeats. Test statistics were conducted and materials were collected at the same time every day. According to the needs of the experiment, the collected samples were quickly cryopreserved in liquid nitrogen or tested directly. The treatment concentration with the greatest phenotypic difference was selected as the working concentration to do further experiments according to the regular statistics of plant height and root length.

### 4.4. Determination of Biomass

The dry weight was determined, and the materials were collected at 0 days and 7 days, respectively, and then dried at 65 °C (Oven, DEPU, China). The effects of aluminum stress on the biomass of *N. cadamba* were expressed by the percentage of aboveground dry weight (ADW)/total plant dry weight (TDW) and belowground dry weight (UDW)/total plant dry weight (TDW), respectively.

### 4.5. Observation on Lignification of Root and Stem

The effects of aluminum stress on the xylem of *N. cadamba* were determined. The third stem node (from top to bottom) and 1 cm root tip of *N. cadamba* in the treatment group and control group were used as experimental materials at 18 d. After embedded in 3% agarose (Biowest, BY-R0100, Spain), the thickness of cross section of stem node was adjusted to 45 μm, the thickness of root cross section was set to 80 μm (vibratome, LeicaVT1000S, Leica Microsystems, Buffalo Grove, IL, USA). Specimens were stained with 0.01% toluidine blue [94] and photos were taken with automatic digital scanning imaging system.

### 4.6. Determination of Root Growth

The root growth after aluminum stress was determined. The whole root system was scanned with WinRHIZO at different time periods (Scanning device LC4800-II, Canada), and then statistical analysis was carried out.

### 4.7. Absorption and Distribution of Aluminum by Root Tip

Eriochrome cyanine R as a chromogenic agent of aluminum, has stability and selectivity, and can form a red-blue complex with aluminum under neutral conditions [95], so the root tips of 1–2 cm cut and treated for different times are cleaned with pure water for 0.5 h, then stained with 0.1% Eriochrome cyanine R solution for 0.5 h, and finally washed with pure water for 0.5 h, observed and photographed under a microscope (XSp-36-1600X-LED, Phoenix, China).

### 4.8. Determination of Chlorophyll

According to the absorption of visible spectrum by chloroplast pigment extract, the extinction of chloroplast pigment was determined by spectrophotometer (UV-1200, MAPADA, China) at a specific wavelength. First, 50 mg of fresh leaves was taken with a punch (except the main veins), then 95% ethanol was added to fix the volume to 5 mL, and the chlorophyll was extracted by shaking. The extracted solution was collected and diluted at a ratio of 1:1, and the extinction was determined at the wavelengths of 470 nm, 649 nm, and 665 nm. The content of chlorophyll is calculated according to the following formula.
(1)$$\mathrm{Chlorophyll}\;\mathrm{content}\;\left(\mathrm{mg}/g\right)\;=\frac{\;\left(\mathrm{concentration}\;\times \;\mathrm{extract}\;\mathrm{volume}\;\times \;\mathrm{dilution}\;\mathrm{multiple}\right)}{\mathrm{fresh}\;\mathrm{weight}\;\mathrm{of}\;\mathrm{the}\;\mathrm{sample}}$$
(2)Chla=13.95D665−6.88D649
(3)Chlb=24.96D649−7.32D665
(4)$$\mathrm{Cx}.c=\frac{1000D470-2.05\mathrm{Ca}-114.8\mathrm{Cb}}{245}$$

Unit: mg/L

The concentration of Chla and Chlb represent chlorophyll A and B in the formula, respectively; the total concentration of Cx.c represents total carotenoids; and D470, D649, and D665 were chloroplast pigment extract under an extinction wavelength of 470nm, 649nm, and 665nm, respectively.

### 4.9. Determination of ROS in Leaves and Roots

According to the ratio of tissue quality (g): extract volume (ml) 1:5-10, the fresh weight of leaf and root tip was 0.1g, respectively. After centrifugation with phosphate buffer solution of pH = 7.0 at 4 °C, the supernatant was taken as the follow-up determination sample. Each set of 3 biological repeats contained 5 technical repeats. SOD, CAT, POD, and MDA were determined (Suzhou Keming Biotechnology Co., Ltd. kit) by Microplate reader (Multiskan FC, Thermo fisher scientific, Waltham, MA, USA). The protein concentration was determined (the BCA method) by Microplate reader (Multiskan FC, Thermo fisher scientific, Waltham, MA, USA).

### 4.10. Determination of Element Concentration in Root Tip

The changes of element concentrations in root tips at different time points after aluminum stress were determined. Root tip samples were collected at different time points under aluminum stress, dried at 65 °C, and then ground and weighed 0.1 g samples were digested by STARTD microwave digestion system (SN131825, Milestoone, Milan, Italy). The microwave digestion tube was added together with 5 mL HNO_3_ and 2 mL 30% H_2_O_2_, and the temperature was raised to 130 °C in 10 min, kept at 130 °C for 10 min, increased to 180 °C within 6min, kept at 180 °C for 30 min, and finally cooled down 30 min in the fume cupboard. After microwave digestion, the mixture was moved to a volumetric flask and fixed volume to 50 mL, then filtered with quantitative filter paper. After the filtered reactants were diluted 100 times, the concentrations of Mn, Fe, Ca, Mg, and K were determined by atomic absorption spectrometer, and the concentrations of Al, P, and S were determined by inductively coupled plasma atomic emission spectrometer (ICP-AES, AIRIS/AP, TJA, Thermo). The calculation formula is
(5)$$\mathrm{Element}\;\mathrm{concentration}\;=\;\frac{\;\left(\mathrm{Csample}\;-\;\mathrm{Cck}\right)\;\mathrm{mg}/L\;\times \;50\mathrm{mL}\times \;\mathrm{dilution}\;\mathrm{multiple}}{\mathrm{dry}\;\mathrm{weight}\;g}$$

Unit: μg/g, mg/g

### 4.11. Construction of Sequencing Library

In this study, total RNA was extracted from 1–2 cm root tips of N. cadamba treated with 0 μM and +Al for 0 days, 1 day, 3 days, and 7 days, and a sequencing library was constructed after qualified detection. A total of 21 libraries (AL0_1, AL0_2, AL0_3, AL1_CK_1, AL1_CK_2, AL1_CK_3, AL1_400_1, AL1_400_2, AL1_400_3, AL3_CK_1, AL3_CK_2, AL3_CK_3, AL3_400_1, AL3_400_2, AL3_400_3, AL7_CK_1, AL7_CK_2, AL7_CK_3, AL7_400_1, AL7_400_2, and AL7_400_3) was constructed with samples from four periods, in which there were three biological repeats in the treatment group and the control group in each period, then further horizontal and vertical analysis. The original sequence (Sequenced Reads) is called clean reads after filtering by Raw reads. The tissue samples of 1–2 cm root tip of *N.cadamba* were collected at different time after aluminum treatment. After quick freezing with liquid nitrogen. RNA, was extracted by Magen (HiPure HP Plant RNA Mini Kit) kit, and then combined with RNA-seq sequencing technology to determine the differentially expressed genes in the root tip of *N.cadamba* after aluminum treatment. The library and sequencing were completed by Nuohe Zhiyuan Bioinformation Technology Co., Ltd. NovoMagic (https://magic.novogene.com/customer/main#/login). TBTools [96] were used for differential gene expression analysis and differential gene enrichment analysis. RNA-seq data were submitted to https://bigd.big.ac.cn/gsub/and the submission number was CRA003281.

### 4.12. qRT-PCR

Use TaKaRa PrimeScript™ RT reagent Kit with gDNA Eraser (Perfect Real Time, Takara, Japan) to synthesize cDNA from the sequenced RNA samples and perform real-time fluorescent quantitative PCR (qRT-PCR) on a LightCycler 480 instrument. The reaction program is 95 °C, 3s; 95 °C, 5s, 40 cycles; 60 °C, 30s [97]. The primers used to determine the transcripts of genes are shown in Appendix A. The reference gene was *N. cadamba* SAMDC [98]. Results were from three biological replicates with 4 technical replicates. Using OriginPro 9.1 64Bit software, the relative gene expression was calculated by the 2^-ΔΔCt^ method [99].

## 5. Conclusions

The results showed that aluminum stress mainly affected the growth and development of *N. cadamba* roots. The Al^3+^ absorbed by the root was preferentially deposited in the root tip, and then transferred to the upper part to influence the xylem thickening of the stem and the decrease of chlorophyll content in leaves. A large amount of Al^3+^ transfer will affect the absorption of other elements and upregulate the expression of related ion transporters, resulting in the disorder of element absorption mechanism and root nutrition imbalance. When Al^3+^ acted on the cell membrane, the imbalance of ROS level was induced, while ROS as a signal molecule triggered hormone changes and then induced the significant expression of related transcription factor genes. Furthermore, ROS regulate the expression of ALMT, MATE, and ABC transporter family genes, and the aluminum transporter AtALS1 upregulated the expression of similar genes significantly and promoted the secretion of organic acids or accumulated dAL^3+^ in vacuoles to form a detoxification mechanism in response to aluminum stress.

## Figures and Tables

**Figure 1 ijms-21-09624-f001:**
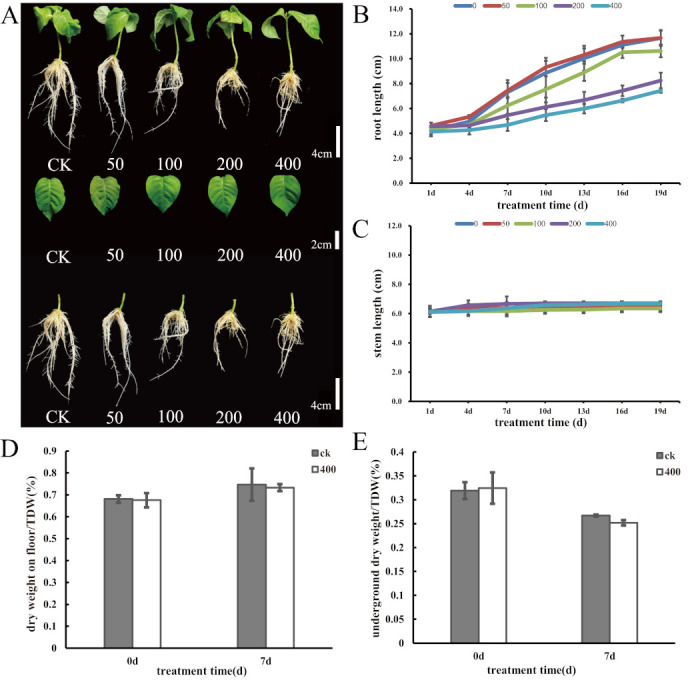
Effects of different concentrations of AlCl_3_ stress. (**A**) The comparison of plant, leaf, and root traits of 0 μM (CK), 50 μM, 100 μM, 200 μM, and 400 μM AlCl_3_ treatment for 24days. (**B**) Changing trend of root length under different concentrations of AlCl_3_ stress. (**C**) Changing trend of plant height under different concentrations of AlCl_3_ stress. (**D**) Changes of aboveground biomass under AlCl_3_ treatment for 0 days and 7 days. (**E**) Changes of belowground biomass under AlCl_3_ treatment for 0 days and 7 days. Results are mean ± SD of three biological replicates. ck-control. 0–0 μM/L, 50–50 μM/L, 100–100 μM/L, 200–200 μM/L, 400–400 μM/L.

**Figure 2 ijms-21-09624-f002:**
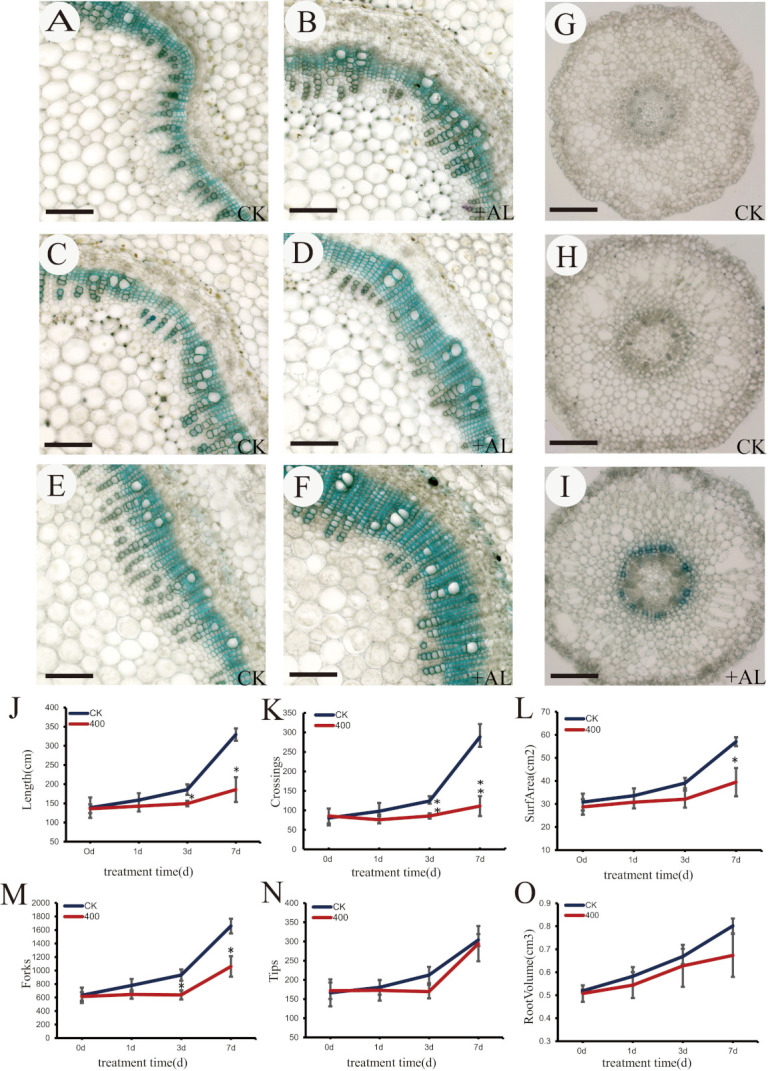
Aluminum stress inhibits the growth of stems and roots. (**A**,**B**) Stem section of control group and treatment group at 1 day. (**C**,**D**) Stem section of control group and treatment group at 3 days. (**E**,**F**) Stem section of control group and treatment group at 7 days. (**G**) Root tip section of control group at 1 day. (**H**) Root tip section of control group at 7 days. (**I**) Root tip section of treatment group at 7 days. The scale bar is 200 μm. (**J**) Total root length. (**K**) Cross number. (**L**) Surface area. (**M**) Branch root number. (**N**) Root tip number. (**O**) Total root volume. The error line represents the average ±SD of the three organisms. The * above the column chart indicated that the growth change between the two samples was significant (Students’ *t*-test,* *p* < 0.05, ** *p* < 0.01).

**Figure 3 ijms-21-09624-f003:**
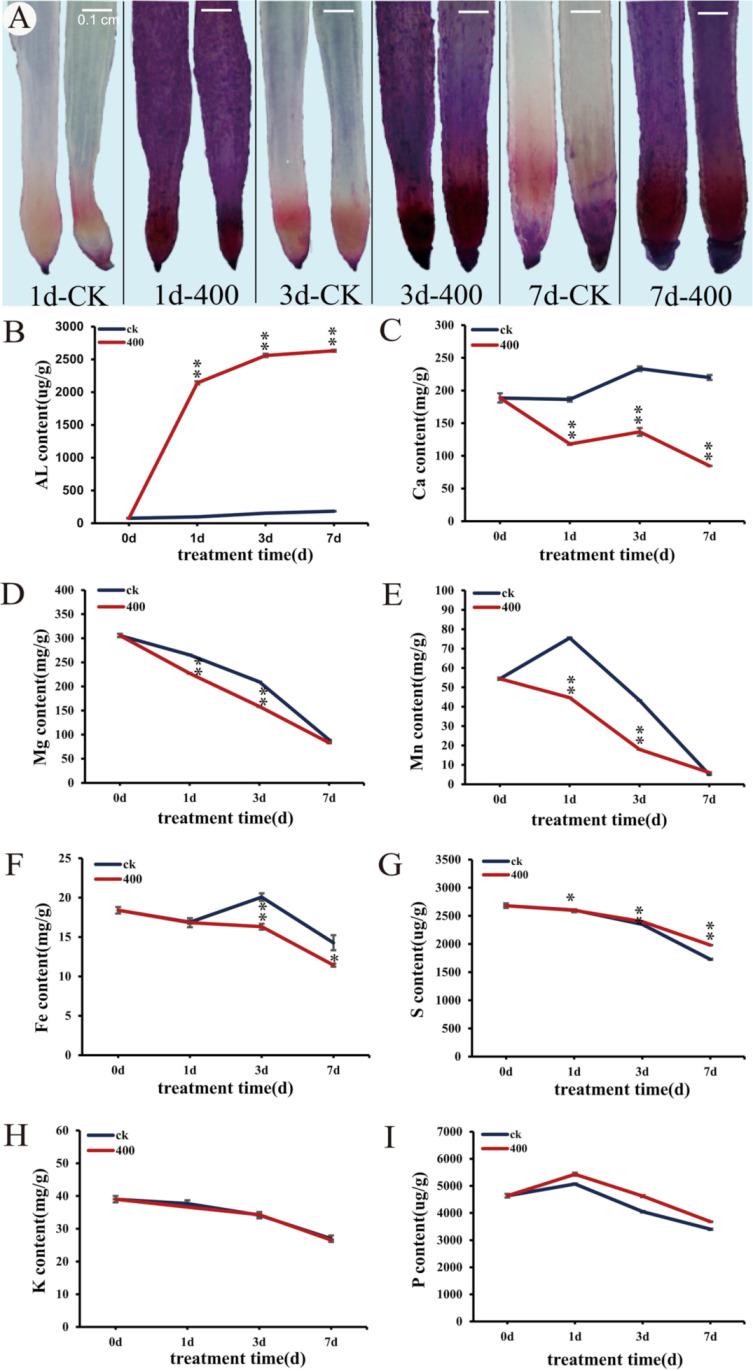
Absorption and distribution of aluminum by root tip. (**A**) Distribution of aluminum in root tip in different periods; the length of the root tip is 1 cm. Eriochrome cyanine R staining does not show red and brown as the control group, but shows red and blue as the treatment group. (**B**-**I**) Changes of element concentration in root tip. The error line represents the average ± SD of the three organisms (Students’ *t*-test, * *p* < 0.05, ** *p* < 0.01).

**Figure 4 ijms-21-09624-f004:**
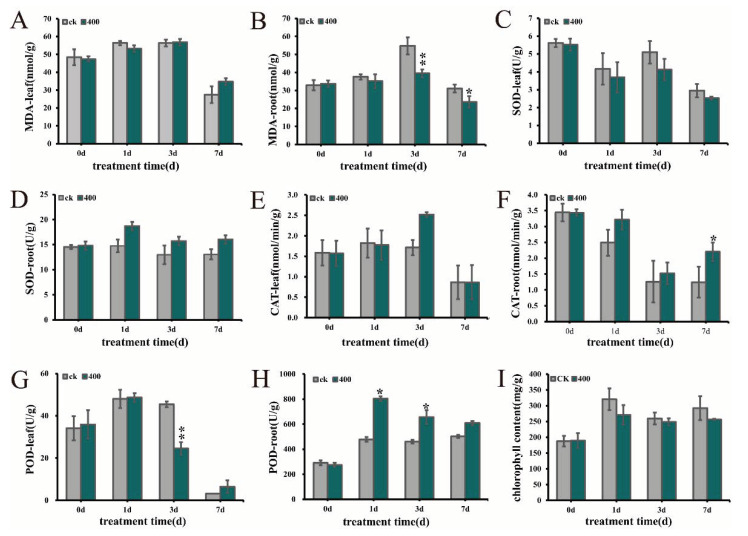
Effects of aluminum stress on ROS and chlorophyll. (**A**–**H**) Changes of ROS in leaves and root tips. (**I**) Chlorophyll content change. The error line represents the average ± SD of the three organisms. The * above the column chart shows that the content of elements between the two samples was significant (Students’ *t*-test, * *p* < 0.05, ** *p* < 0.01).

**Figure 5 ijms-21-09624-f005:**
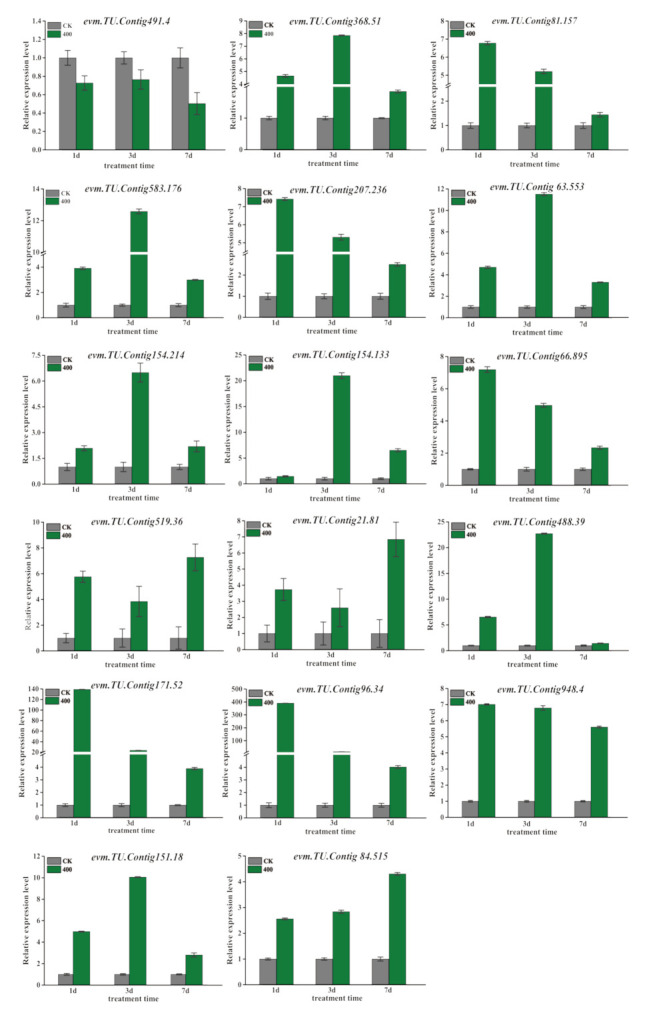
qRT-PCR verifies the relative expression levels of differential expression genes (DEGs) at different times under aluminum stress. Transport (*evm.TU.Contig488.39, evm.TU.Contig519.36, evm.TU.Contig154.214, evm.TU.Contig154.133, evm.TU.Contig491.4, evm.TU.Contig63.553*); transcription factor (*evm.TU.Contig96.34, evm.TU.Contig948.4, evm.TU.Contig151.18, evm.TU.Contig84.515*); protein (*evm.TU.Contig368.51, evm.TU.Contig81.157, evm.TU.Contig 171.52, evm.TU.Contig583.176, evm.TU.Contig207.236, evm.TU.Contig66.895, evm.TU.Contig 21.81*).

**Figure 6 ijms-21-09624-f006:**
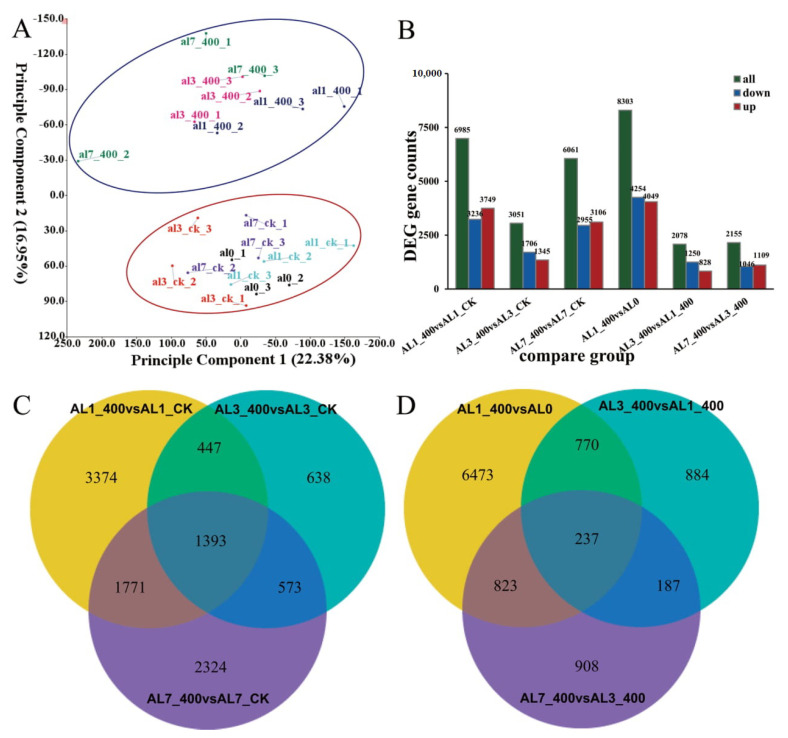
The DEGs of *N.cadamba* in response to aluminum stress. (**A**) Principal component analysis (PCA) of DEGs: diagram of treatment group and control group at 0 days, 1 day, 3 days, 7 days, the dots with the same color represent 3 biological replicates of the same treatment group, the blue area and the red area are the treatment group and control group gene cluster, respectively. (**B**) The green bar represents all DEGs, downregulated DEGs are in blue, and upregulated DEGs are in red, FDR < 0.05. (**C**,**D**) The overlapping area represents the DEGs with a common regulation mode between different treatments.

**Figure 7 ijms-21-09624-f007:**
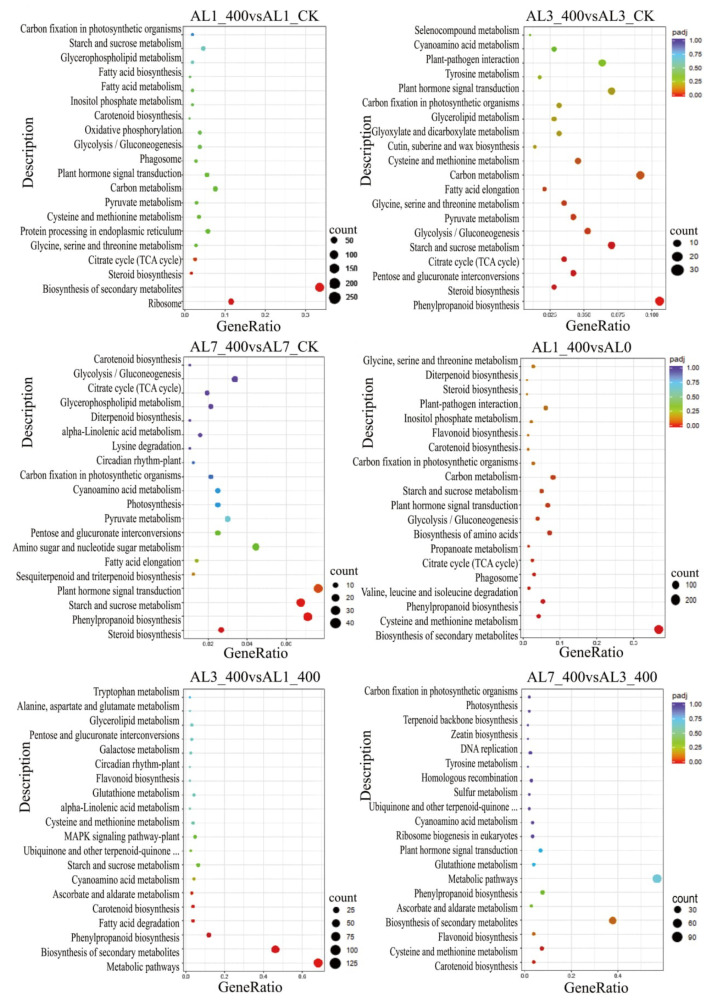
KEGG (Kyoto Encyclopedia of Genes and Genomes) enrichment of six comparative combinations of aluminum stress in different periods. The ordinate indicates the name of the pathway, the abscissa indicates the rich factor, the size of the dot indicates the number of differentially expressed genes in the pathway, and the color of the dot corresponds to different q value ranges.

**Figure 8 ijms-21-09624-f008:**
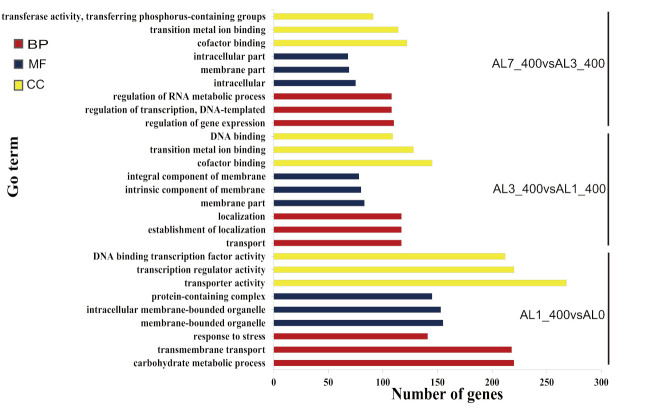
Differential gene GO enrichment among different combinations of aluminum stress comparison. The abscissa is the GO term, the enrichment is the number of differential genes in the term, and 27 GO terms with the most significant enrichment in different periods are selected to show in the map. The red bar column represents the DEGs which enriched on biological process; the blue bar column represents the DEGs which enriched on molecular function; the yellow bar column represents DEGs which enriched on cellular component.

**Figure 9 ijms-21-09624-f009:**
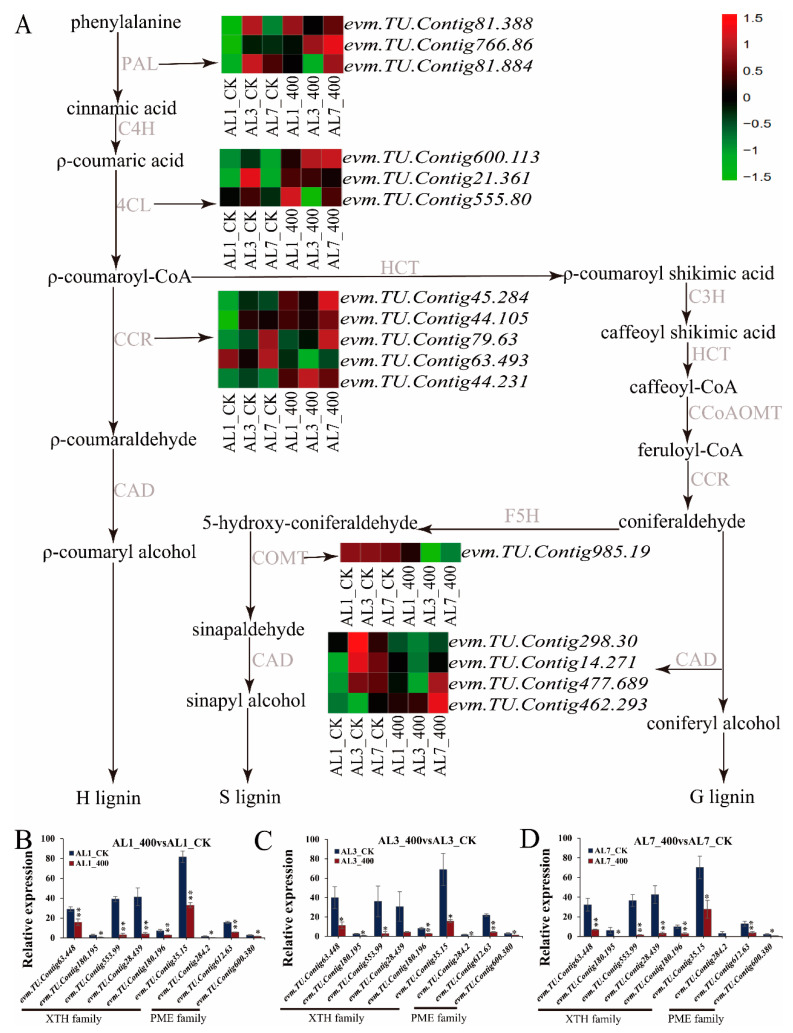
Expression of cell wall-related genes induced by aluminum stress. (**A**) Enrichment of DEGs in the phenylalanine pathway. (**B**–**D**) XTH/PME family DEGs relative expression. FDR < 0.05, |log_2_(FoldChange)| > 1. The * above the column chart showed that the electrical conductivity changed significantly between the two samples (* *p* < 0.05, ** *p* < 0.01).

**Figure 10 ijms-21-09624-f010:**
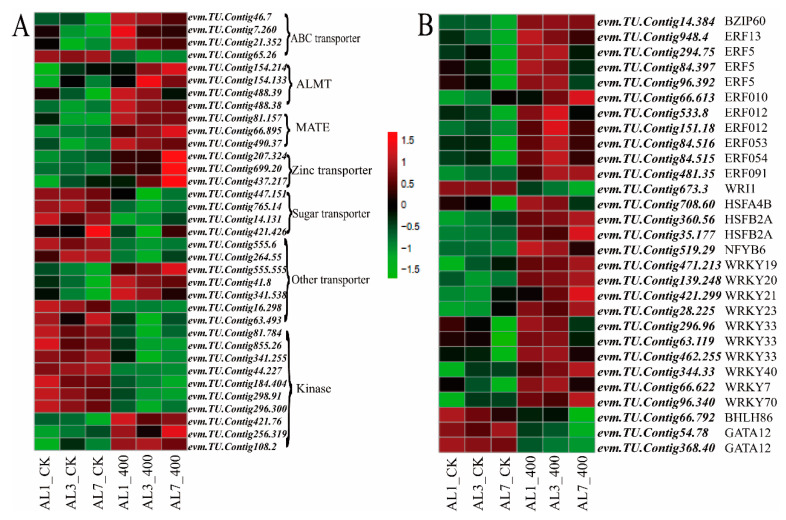
Expression of related transporters and transcription factors induced by aluminum stress. The threshold of FDR < 0.05, |log2 (FoldChange)| > 1 was used to screen the changes of differential gene expression between the treated group and the control group after aluminum stress at different times. (**A**) Heat map of aluminum transport-related gene and related protein kinase expression induced by aluminum stress. (**B**) Heat map of transcription factor-related gene expression induced by aluminum stress.

**Figure 11 ijms-21-09624-f011:**
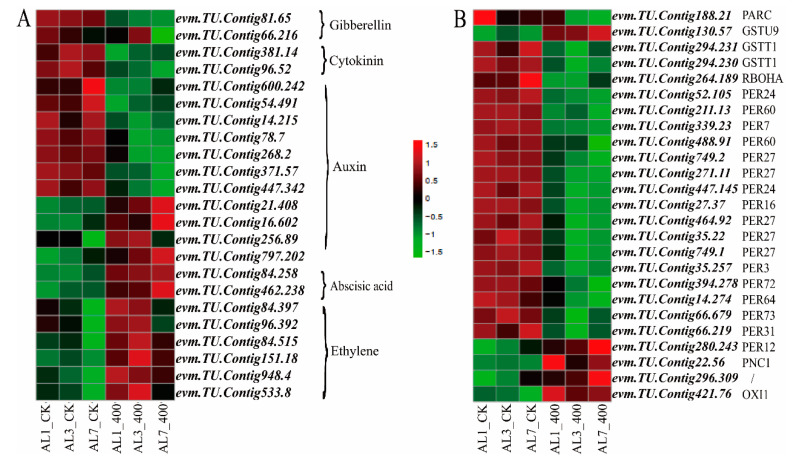
Aluminum stress induces the expression of ROS and hormone-related genes. The threshold of FDR < 0.05, |log2 (FoldChange)| > 1 was used to screen DEGs between the treated group and the control group after aluminum stress at different times. (**A**) Heat map of hormone-related gene expression. (**B**) Heat map of ROS-related gene expression.

**Figure 12 ijms-21-09624-f012:**
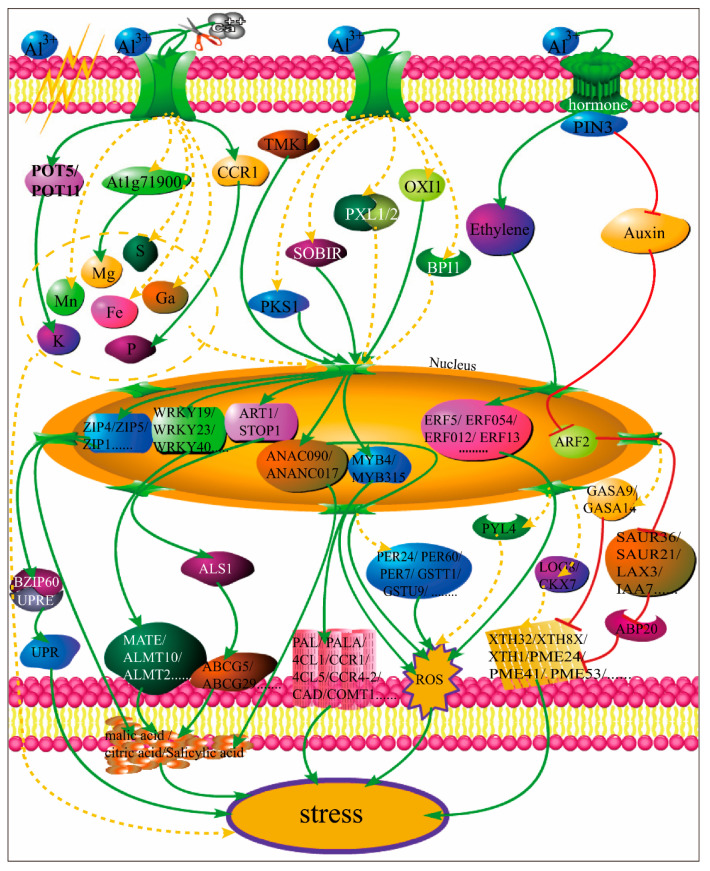
Aluminum stress pathway diagram at the cell level in *N. cadamba*. The green solid line indicates that the upstream gene regulation activates the downstream gene, the red solid line indicates that the upstream gene regulation suppresses the downstream gene, and the yellow dotted line indicates that the upstream gene indirectly or possibly regulates the downstream gene and responds to stress response.

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
