# Peer review of "Physiological, Biochemical, and Transcriptomic Responses of Neolamarckia cadamba to Aluminum Stress"

_ijms, 2020, doi:10.3390/ijms21249624_

Round 1

Reviewer 1 Report

The manuscript describes a significant amount of research work. The group studied physiological, biochemical and transcriptomic responses of Neolamarckia cadamba to aluminum stress. It is an interesting attempt to study different aspects, although the results are the more or the less confirmatory and have been described in other experimental systems. In order to highlight the significance of their work, the authors should focus on novel findings and describe them very clear in the conclusions. 

Some specific suggestions

  1. The work describes physiological, biochemical and transcriptomic responses of Neolamarckia cadamba to aluminum stress and their interconnection. Title is misleading because it seems that transcriptomic analysis provided the lead for physiological and biochemical targeted studies. A suggestion to consider for the title is: "Physiological, biochemical and transcriptomic responses of Neolamarckia cadamba to aluminum stress"
  2. Description of transcriptomic analysis design (lines 622-628) and related results need significant revision. To me it is unclear what was compared to what. The control of the experiment (CK) in line 132 is decribed as treatment (line 140) and in M&M line 622 "treated with CK". 
  3. Provide details of sampling in 4.3 (lines 557-562). When did you get samples (was the same time each day?) How did you store samples etc.
  4. Revise the conclusions highlighting significant novel findings of the study

Overall, the paper describes a large amount of work to decipher aluminum responses of  Neolamarckia cadamba and it is worthy of publication. A minor revision could improve the manuscript and is highly recommended.

Author Response

List of Responses

Dear Editor:

Thank you for your letter and for the reviewers’ comments concerning our manuscript entitled “Transcriptome analysis revealed Physiological and Molecular Level Responses to Aluminum stress in Neolamarckia cadamba”. Those comments are all valuable and very helpful for revising and improving our paper, as well as the important guiding significance to our researches. We have studied comments carefully and have made correction which we hope meet with approval. Revised portion are marked in blue in the paper. The main corrections in the paper and the responds to the reviewer’s comments are as flowing:

Reviewer 1

Responds to the reviewer’s comments: 

Some specific suggestions

  1. The work describes physiological, biochemical and transcriptomic responses of Neolamarckia cadambato aluminum stress and their interconnection. Title is misleading because it seems that transcriptomic analysis provided the lead for physiological and biochemical targeted studies. A suggestion to consider for the title is: "Physiological, biochemical and transcriptomic responses of Neolamarckia cadamba to aluminum stress"

Response: 

Thank you very much for the evaluator’s suggestion. Indeed, as suggested by the reviewer, this title is more in line with the content of our article, and we agree with your suggestion, and we can easily put forward your suggestions, so after our discussion, the title was replaced with " Physiological, biochemical and transcriptomic responses of Neolamarckia cadamba to aluminum stress "(line 2-4).

  1. Description of transcriptomic analysis design (lines 622-628) and related results need significant revision. To me it is unclear what was compared to what. The control of the experiment (CK) in line 132 is decribed as treatment (line 140) and in M&M line 622 "treated with CK". 

Response:

Thank you very much for the reviewer’s suggestion. I’m very sorry that we didn’t clearly indicate the combination of our experiment here (lines 622-628), so we have re-written this part according to the Reviewer’s suggestion (lines 626-630). In the experiment, we measured a total of 21 samples, which included 4 periods (0d, 1d, 3d, 7d). Among them, the samples of 0d were not often treated with Al stress, so only 3 biological replicates were used as controls, while the samples of 1d, 3d and 7d included treatment group and control group. Each period includes the treatment group and the control group, and each group contains 3 biological replicates. In addition, our subsequent analysis includes horizontal comparison and vertical comparison. The horizontal comparison is mainly based on AL1_400 vs AL1_CK, AL3_400 vs AL3_CK, AL7_400 vs AL7_CK, and the three comparison combinations are analyzed, while the vertical comparison is mainly based on AL1_400 vs AL0, AL3_400 vs AL1_400, AL7_400 vs AL3_400 are three comparative combinations for a simple analysis.

I'm very sorry to write the control group here again as treatment (line132,140,622), thank you very much for the reviewers to find the problem in time, we also made corresponding changes to this part of the problem according to your suggestions (line 136,140,624)

  1. Provide details of sampling in 4.3 (lines 557-562). When did you get samples (was the same time each day?) How did you store samples etc.

Response:

Thank you very much for the reviewer’s advice. We are divided into four periods for measuring and collecting materials. As you mentioned, the materials are collected at the same time every day to ensure the accuracy of our time. So I have added the details of sampling and saving samples according to your suggestion (lines 560-562).

  1. Revise the conclusions highlighting significant novel findings of the study

Response:

Thank you very much for the reviewer’s advice. We have modified this part of the content to a certain extent based on your valuable suggestions to highlight the highlights and key points of our article (line 650-655).

Special thanks to you for your good comments.

We tried our best to improve the manuscript and made some changes in the manuscript. These changes will not influence the content and framework of the paper. And here we list all the changes and marked in yellow in revised paper. We appreciate for Editors/Reviewers’ warm work earnestly, and hope that the correction will meet with approval.

Once again, thank you very much for your comments and suggestions

Reviewer 2 Report

The authors explore the effects of Al+3 on a tree species with economic uses from tropical/subtropical areas. In addition to analyses of plant growth and element concentration, they also explore gene expression as affected by Al+3 treatments. Their findings are integrated with our knowledge of Al+3 impacts on other economically important herbaceous plants, as well as Arabidopsis.

The research appears well designed and executed. I say “appears” because the English writing is sometimes difficult to understand and there is a lot of detail presented in the figures. But overall the study examines a large number of factors and so provides a comprehensive study of this species.

The manuscript would benefit from careful copy-editing. There were many places where the grammar was incorrect or there were other problems caused by translation to English. As an example, I am uploading a pdf in which I take several lines of the original manuscript and then edit them to illustrate a number of errors.

Another example of the copy-editing needed can be seen in Figure 1, where in panel D the x-axis label reads “dry weight on the floor” but I think “dry weight belowground” was intended. Note also that the legend for the figure uses “Underground” and so is closer to the best term (yet not quite there). As a final example, in Figure 2 panel M the x-axis label is “forks” but I think “branch root number” is intended. Also in the legend for Fig. 2 there is an error at the very end: “* p < 0.01, ** p < 0.01.” I suspect the two asterisks signify p < 0.001. These are simply examples of many other errors that need to be identified and corrected by careful copy-editing.

There are a couple other problems that should be addressed. First, the authors use the word “content” when they should be using “concentration.” All values in the manuscript reported as ug/g are actually concentrations. Content would be the total mass of an element in a structure. This change is needed in the text but also in the figure axis labels and figure legends.

 Second, the authors should be mindful of significant digits when reporting elemental concentrations. As an example, on line 194 we read a mean Al concentration of 2141.441 ug/g was reached at day 0. Using 7 significant digits, with 3 decimal places, is very much not necessary and implies an unrealistic precision in measurement of concentration in the samples. More examples are found through line 206 in that section. I’d suggest 3 significant digits as sufficient.

Finally, in Lines 142-144, the authors report that aboveground and belowground dry weight decreased in Fig. 1. Was the decrease statistically significant? No statistical analysis of the data is provided in this case.

Author Response

List of Responses

Dear Editor:

Thank you for your letter and for the reviewers’ comments concerning our manuscript entitled “Transcriptome analysis revealed Physiological and Molecular Level Responses to Aluminum stress in Neolamarckia cadamba”. Those comments are all valuable and very helpful for revising and improving our paper, as well as the important guiding significance to our researches. We have studied comments carefully and have made correction which we hope meet with approval. Revised portion are marked in blue in the paper. The main corrections in the paper and the responds to the reviewer’s comments are as flowing:

Reviewer 2

  •  

Comments and Suggestions for Authors

  1. The authors explore the effects of Al+3 on a tree species with economic uses from tropical/subtropical areas. In addition to analyses of plant growth and element concentration, they also explore gene expression as affected by Al+3 treatments. Their findings are integrated with our knowledge of Al+3 impacts on other economically important herbaceous plants, as well as Arabidopsis.
  2. The research appears well designed and executed. I say “appears” because the English writing is sometimes difficult to understand and there is a lot of detail presented in the figures. But overall the study examines a large number of factors and so provides a comprehensive study of this species.

Response:

Thank you very much for the reviewer’s advice. We have modified this part of the content to a certain extent based on your valuable suggestions (line 21-33,44-57,100-105,133-135,178,161-164,200-209,217-224,344-346,353,491,519,534).

  1. The manuscript would benefit from careful copy-editing. There were many places where the grammar was incorrect or there were other problems caused by translation to English. As an example, I am uploading a pdf in which I take several lines of the original manuscript and then edit them to illustrate a number of errors.

Response:

I'm very sorry for the language error in the article, and thank the reviewer for meticulousness. We have re-made relevant changes based on the PDF file you uploaded (lines 48-57).

  1. Another example of the copy-editing needed can be seen in Figure 1, where in panel D the x-axis label reads “dry weight on the floor” but I think “dry weight belowground” was intended. Note also that the legend for the figure uses “Underground” and so is closer to the best term (yet not quite there). As a final example, in Figure 2 panel M the x-axis label is “forks” but I think “branch root number” is intended. Also in the legend for Fig. 2 there is an error at the very end: “* p < 0.01, ** p < 0.01.” I suspect the two asterisks signify p < 0.001. These are simply examples of many other errors that need to be identified and corrected by careful copy-editing.

Response:

Thank you very much for the reviewer’s suggestion. We are very much agree with your suggestion. We have revised the corresponding diagram and related terms in the article based on your valuable suggestions (lines 143-144,154-155,569, Figure 1; line 169,178, Figure 2).

We are very sorry for the error of "* p <0.01, ** p <0.01". Through discussion, we believed that "* p <0.05, ** p <0.01" is statistically significant calculations, so we have corrected and replaced such errors in the article (lines 181,208,215,233,367), because our statistics are also calculated according to this standard.

  1. There are a couple other problems that should be addressed. First, the authors use the word “content” when they should be using “concentration.” All values in the manuscript reported as ug/g are actually concentrations. Content would be the total mass of an element in a structure. This change is needed in the text but also in the figure axis labels and figure legends.

Response:

We are very sorry for the mistake of improper wording in the article. We also very much agree that ug/g is the concentration of each element, so we have corrected the element concentration in the article based on your valuable suggestions (line 25-28,185-207,214, 386-387,493,496,503,508,608,609,621, Figure 3).

  1. Second, the authors should be mindful of significant digits when reporting elemental concentrations. As an example, on line 194 we read a mean Al concentration of 2141.441 ug/g was reached at day 0. Using 7 significant digits, with 3 decimal places, is very much not necessary and implies an unrealistic precision in measurement of concentration in the samples. More examples are found through line 206 in that section. I’d suggest 3 significant digits as sufficient.

Response:

Thank you very much for the good suggestion of the reviewer. We agree with your suggestions. After our discussion, we decided to keep the element concentration as an integer, which is closer to the actual accuracy. The revised part in the article includes line 194-207.

  1. Finally, in Lines 142-144, the authors report that aboveground and belowground dry weight decreased in Fig. 1. Was the decrease statistically significant? No statistical analysis of the data is provided in this case.

Response:

Thank you very much for the valuable suggestions of the reviewers. In Figure 1, we have detected that the aboveground and belowground biomass has indeed been reduced, but due to the short-term effect of treatment time, the aboveground and belowground biomass that we have detected have decreased slightly, and there is no statistical significance, so in this case we did not provide statistical analysis of the data. But we also made a brief explanation in the article based on your suggestion (line 143-144).

Special thanks to you for your good comments.

We tried our best to improve the manuscript and made some changes in the manuscript. These changes will not influence the content and framework of the paper. And here we list all the changes and marked in yellow in revised paper. We appreciate for Editors/Reviewers’ warm work earnestly, and hope that the correction will meet with approval.

Once again, thank you very much for your comments and suggestions
